# Quantitative risk assessment and interventional recommendations for preventing canine distemper virus infection in captive tigers at selected wildlife stations in Thailand

Kanittha Tonchiangsai[1,2], Anuwat Wiratsudakul[1,2], Suwicha Kasemsuwan[3], Ruangrat Buddhirongawatr[1,2], Weerapong Thanapongtharm[4], Kan Kledmanee[5], Tatiyanuch Chamsai[2], Nareerat Sangkachai[2], Bencharong Sangkharak[6], Pakpoom Aramsirirujiwet[6], Sarin Suwanpakdee[1,2]*

1 Department of Clinical Sciences and Public Health, Faculty of Veterinary Science, Mahidol University, Phutthamonthon, Nakhon Pathom, Thailand, 2 The Monitoring and Surveillance Center for Zoonotic Diseases in Wildlife and Exotic Animals, Faculty of Veterinary Science, Mahidol University, Phutthamonthon, Nakhon Pathom, Thailand, 3 Department of Veterinary Public Health, Faculty of Veterinary Medicine, Kasetsart University, Kampangsan, Nakhon Pathom, Thailand, 4 Bureau of Disease Control and Veterinary Services, Department of Livestock Development, Ratchathewi, Bangkok, Thailand, 5 Department of Epidemiology, Faculty of Public Health, Mahidol University, Ratchathewi, Bangkok, Thailand, 6 Wildlife Conservation Office, Department of National Parks, Wildlife and Plant Conservation, Chatuchak, Bangkok, Thailand

* sarin.suw@mahidol.edu, sarin.suw@mahidol.ac.th

## Abstract

Canine distemper virus (CDV) can cause high morbidity and mortality in large felids and pose a significant threat to the conservation of captive and non-captive tiger (*Panthera tigris*). This study conducted in Thailand's wildlife stations aimed to assess the risks of CDV introduction to captive tiger populations. A stochastic quantitative risk assessment model was employed to determine the pathways and estimate the risk probabilities through humans, animal reservoirs, and fomites. The final risk probability of entry, obtained from a combination of six entry pathways, indicated that the absence of measures resulted in a relatively high risk at 0.858. The sensitivity analysis identified CDV-contaminated human hands, followed by other CDV-infected wild animals, and CDV-contaminated equipment, as the most influential pathways of CDV spread. Risk probabilities were compared among those without intervention, with routine intervention at wildlife stations, and with full intervention implementation. Implementing all interventions at the captive wildlife stations significantly reduced the risk of CDV introduction. These interventions included control measures such as quarantining and isolating infected animals and providing treatment to reduce infectiousness. Preventive measures included screening tests for healthy individuals for early detection of asymptomatic or pre-symptomatic cases, preventing further spread or complications, CDV vaccination campaigns, and promoting hand hygiene among staff and visitors. Environmental interventions involve restricting dogs and cats from accessing tiger enclosures, disinfecting animal transport vehicles, using

provided the original author and source are credited.

**Data availability statement:** All relevant data are within the manuscript and its Supporting Information files.

**Funding:** The author(s) received no specific funding for this work.

**Competing interests:** The authors have declared that no competing interests exist.

separate equipment for each cage, etc. Together, these interventions lowered the median risk of CDV introduction to 0.089, representing an 89.6% risk reduction. This approach assessed CDV infection risks and adapted interventions to specific situations at wildlife stations. Consistent implementation of these measures is essential to minimize CDV spread. Wildlife stations must strictly implement these interventions as standard procedures to protect the health of captive tigers.

## Introduction

Canine distemper virus (CDV) is an enveloped single-stranded RNA virus of the family *Paramyxoviridae* in the genus *Morbillivirus* [1]. Distemper is a highly contagious and potentially fatal disease that attacks a dog's respiratory, gastrointestinal, and nervous systems. CDV is spread through the air by infected coughs or sneezes and could be transmitted through contact with bodily fluids [2]. Although puppies and unvaccinated dogs are most at risk, all dogs are susceptible to the virus [2]. It can also infect other animals within the order Carnivora, including foxes (*Vulpes vulpes*), raccoons (*Procyon lotor*), bears (*Ursus* spp.), and large felids (*Panthera* spp.) and species outside this order, such as the rhesus monkey (*Macaca mulata*), collared anteater (*Tamandua tetradactyla*), and Asian elephant (*Elephas maximus*) [2]. Clinical symptoms in infected tigers and other carnivores range from asymptomatic to nasal discharge, conjunctivitis, hyperkeratosis at footpads and nasal planum, diarrhea, melena, vomiting, seizure, opisthotonos, and myoclonus [3–6]. Laryngeal paralysis and stridor sounds were also observed in infected tigers [5].

Tigers (*Panthera tigris*) are an endangered species [7]. Thailand has approximately 190-250 wild tigers, mainly found in western forests [8]. Meanwhile, captive tigers are housed in wildlife stations and zoos and by private owners [9–11]. The government has implemented strict laws to protect tigers in Thailand [12]. The Department of National Parks, Wildlife, and Plant Conservation (DNP) oversees wildlife breeding and rescue centers (referred to as wildlife stations in this study), and the Zoological Park Organization of Thailand (ZPOT) manages governmental zoos. As of 2021, Thailand has a total of 164 captive tigers: 119 in DNP wildlife stations (S1 Table in S1 File) and 45 in ZPOT zoos [10]. Tigers are kept in legal captivity for conservation. Wildlife stations participate in captive rescues as part of ex situ conservation efforts. Health and disease prevention management is crucial to maintain the health of captive tigers.

In tropical and subtropical ecosystems, tigers serve as an umbrella and iconic species, and their conservation supports broader biodiversity by maintaining ecological balance and protecting coexisting species [13]. As apex predators, they regulate prey populations, contributing to their ecosystem health [14]. However, wild tigers are facing an additional threat from CDV, which is associated with illness and death in large felids [15]. CDV poses a potential threat to the extinction of Amur tigers (*Panthera tigris altaica*) with small populations, particularly those in Russia. A significant CDV outbreak affected Amur tigers in 2004 and 2010 within the Sikhote-Alin Biosphere Reserve in Russia, leading to a dramatic population decline from 25 individuals in 2008 to only 9 in 2012. Infected tigers exhibited abnormal behaviors, such as fearlessness and neurological symptoms, which caused human wildlife conflicts and fatalities. This outbreak highlighted the susceptibility of small tiger populations to diseases introduced from reservoirs of more abundant species, such as domestic dogs and wild carnivores [16].

In 2011, 12 captive tigers at a Japanese zoo contracted respiratory and gastrointestinal diseases. A tiger died from neurological complications, and CDV was found in its fecal samples

and lungs. This outbreak may have been spread by wildlife around the zoo [4]. In 2021, a CDV outbreak resulted in the death of seven captive tigers and one lion in an exotic feline rescue center in the USA. It was linked to being near or sharing enclosures with infected animals of the same species [17]. In 2016, two captive wildlife stations in Thailand experienced a major CDV outbreak. Most of the 156 affected tigers were Siberian tigers taken from private collectors. A total of 88 tigers died; the majority of deaths were suspected to be caused by CDV infection, with 31.88% (22 out of 69 dead tested tigers) primarily attributed to CDV infection. Neurological signs were observed after the systemic infection had developed. Other animals, including bears, monkeys, and leopard cats, also tested positive for CDV during this outbreak [5].

Therefore, risk estimation of CDV infection is needed in captive large felids to recommend the best practices for disease prevention. The risk assessment results help in the management of quarantined and sick animals and the prevention of the spread of viral particles into the environment. In this study, stochastic risk assessment of CDV infection for captive tigers in the wildlife stations was used. It is a protocol for determining the risk groups and defining the risk level. Finally, the effectiveness of risk-reducing interventions was assessed by entering them into the risk estimation models. This study aimed to present best practices to mitigate the risk of CDV introduction to captive wildlife stations.

## Materials and methods

### Study sites

Captive tigers are found in 10 of 26 Thailand's DNP wildlife stations, including wildlife rescues and breeding stations. This study focused on 7 stations located in different regions in Thailand, which have more than 2 tigers. Three stations with only 1-2 tigers and potentially different husbandry and management practices were excluded (Fig 1). This figure was generated by the authors using QGIS version 3.30.3-'s-Hertogenbosch, with base map data obtained from the Thai Land Development Department (accessible at https://tswc.ldd.go.th/DownloadGIS/Index_Lu.html, accessed on [May 30, 2024]).

### Data collection

Data were collected from 2016 to 2021 through interviews using semi-structured questionnaires. These questionnaires contained a mix of predefined questions with specific response options (structured parts) alongside open-ended questions that allowed respondents to provide more detailed responses. The interviews were conducted on-site and online. The plan was to interview one director or veterinarian per wildlife station and approximately three animal keepers per station, or as many as were available. Across seven wildlife stations, a total of interviews was conducted. This included interviews with directors from four stations and veterinarians from three stations using questionnaire number 1 (QN1). In addition, due to variations in the number of staff at different stations, a total of 18 interviews with keepers were conducted. These comprised 15 tiger keepers, 2 keepers of other susceptible wild animals (1 bear keeper and 1 monkey keeper), and 1 animal husbandry staff member. All keeper interviews were conducted using QN2. Both QN1 and QN2 are provided in the supporting information (S9 in S1 File). Nevertheless, information from various other sources, such as health and husbandry records of wildlife stations, laboratory databases, published literature, public databases, relevant websites, and standard guidelines, were used as input parameters and probabilities incorporated into the quantitative stochastic models employed in this study. Subsequently, the backbone pathways of CDV entry to wildlife stations were developed based on the collected data.

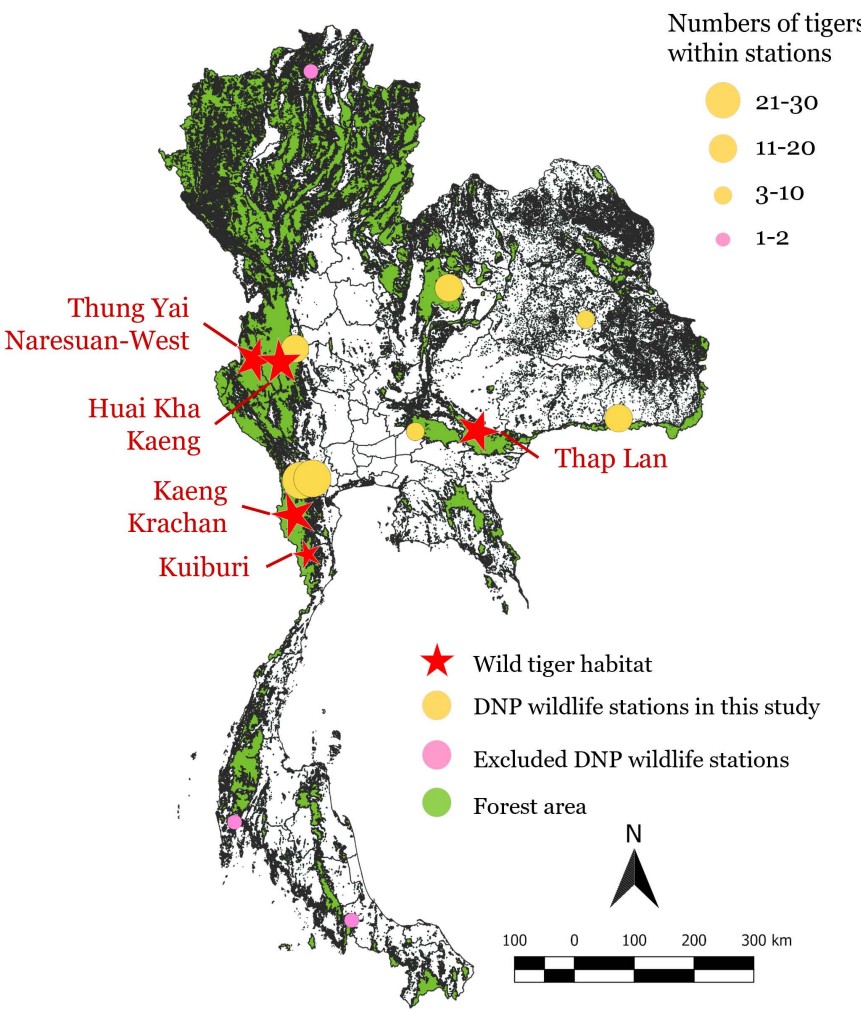

**Fig 1. DNP wildlife stations with captive tigers and wild tiger habitats in Thailand (2021).** Yellow circles show the included stations, whereas pink circles show the excluded ones. The sizes of the circles represent the number of tigers. Red stars indicate wild tiger habitats, and green areas represent forests.

### Data analysis

The quantitative risk assessment model was developed, and calculations were made using the R Program Version 4.3.0 within the interface R-studio Program Version 2023.03.1. Probability distributions were selected with the Monte Carlo sampling method from the package "mc2d" to run entry scenarios with 1,000 iterations. Median probabilities with 5th and 95th percentile values were determined. The sensitivity analysis was conducted to identify the most influential parameters on the model outputs. Risk-mitigated interventions were chosen to analyze risk reduction after identifying the most potential entry pathways.

### Quantitative stochastic risk assessment model

Routes or physical pathways of CDV entry to CDV-free wildlife stations were identified without any interventions (Fig 2). The entry boundaries for all animal infections and fomite contaminations began at the entrance of tiger housing areas within the wildlife stations.

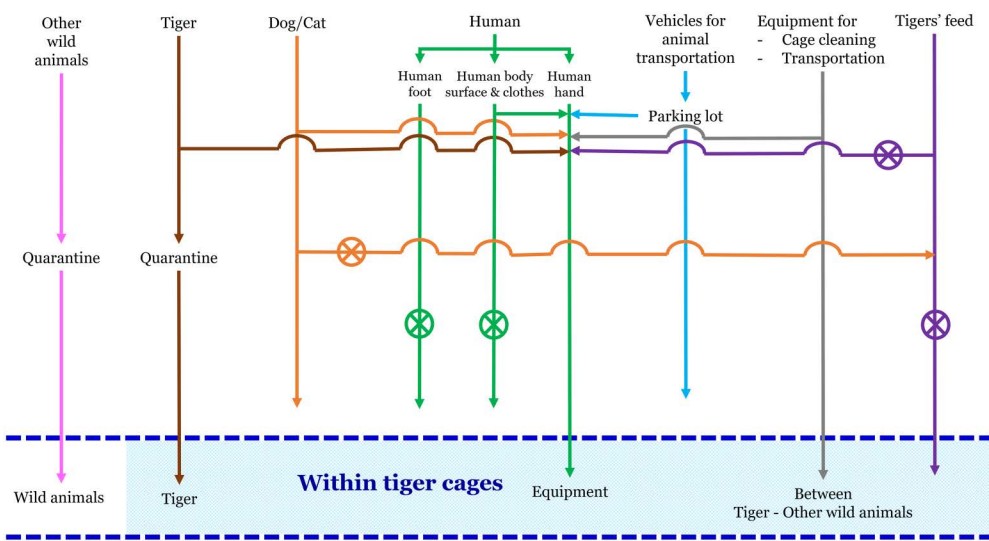

**Fig 2. Backbone layouts of DNP wildlife stations and physical pathways of entry of humans, animals, and fomites to the wildlife stations.** The arrowheads show the directions of entry, and the crosses in circles show conditions without entry in those routes.

Given that not all transmission routes can infect tigers in the housing areas, CDV can enter wildlife stations through infected tigers, other wild animals, domestic dog or cat reservoirs, human contamination (hands, bodies, and feet), animal-transported vehicles, equipment, and tigers' food. The Australian guideline for importing non-domestic Felidae requires diagnostic tests, treatment efficacy information, vaccinations, and appropriate quarantine durations [18]. Following this guideline, tigers and other wild animals need to be tested for CDV to ascertain their disease status before transportation, quarantined for 30 days after arrival to observe symptoms, and tested again for CDV. After testing, sick animals are isolated, treated, and/or given CDV vaccination. The presence of domestic dogs and cats near wildlife stations and their ability to access housing areas of tigers or other wild animals were also considered in the entry pathway because CDV aerosol transmission between animals can occur.

Humans in close contact with tigers, such as staff members, can transmit CDV to tigers or other animals. This occurs when humans touch equipment used in tigers' cages, tigers' food, animal transport vehicles, or their pets at home. Visitors who do not have direct contact with tigers can also contribute to contamination by touching food tongs with their hands. Therefore, human hand contamination was included in the entry pathway. However, human bodies and contaminated footwear were omitted from the entry pathway because visitors are prohibited from entering tiger cages. In addition, staff members responsible for tiger care do not enter the tiger cages directly, as tigers are wild animals with instinctive aggression. Staff members do not make physical contact with the tigers. Cleaning tasks, such as washing floors or cleaning water ponds, are generally performed only after the tigers have been moved to another cage. During the survey of wildlife stations, no tiger cubs were present; thus, contamination through external human body contact, such as hugging or holding them, was not possible.

Wildlife stations have animal transport vehicles, which can be accessed near the tiger cages. CDV can contaminate vehicles carrying infected animals; thus, this study focused only on

exterior surfaces such as wheels or mudguards. In addition, equipment was shared among tiger and other animal cages, such as floor-cleaning brushes used between tiger and small wild cat cages, and cages used to transport tigers were also used to transport monkeys. Domestic dogs could not enter tiger cages or pantries in wildlife stations; thus, the risk of contamination through tigers' food from infected dogs was eliminated. CDV contamination via food handlers or outsiders was also excluded, as food preparation staff members work only in designated areas with strict hygiene protocols, and outsiders never handle the feed. In instances where feeding activities occur at certain wildlife stations, tools are utilized to avoid direct contact.

Thus, the physical pathways of CDV introduction to wildlife stations in Fig 2 contained six biological pathways, namely, i) CDV-infected tigers, ii) other CDV-infected wild animals, iii) dog or cat reservoirs, iv) CDV-contaminated human hands, v) wheels or mudguards of animal transport vehicles, and vi) CDV-contaminated equipment. The biological pathways and sub-equations for transmission through tigers, other wild or domestic animals, human hands, animal transport vehicles, and equipment are presented in S2–S5 Figs in S1 File.

### Entry assessment model specification

To estimate the annual probabilities of CDV introduction to captive tigers within Thailand's DNP captive wildlife stations through animal infections and fomite contaminations, six quantitative stochastic risk assessment models were developed. Each probability of entry ($P_{Ri}$) of six pathways through which CDV can infect or contaminate from the outer to destined wildlife stations annually was modeled as the products of various conditional probabilities as follows:

$$P_{R_i} = \sum_{i=1}^{n_i} R_i \Big/ \left( \sum_{i=1}^{n_i} R_i + \sum_{i=1}^{n_i} N_j \right)$$

$$; R_i = \prod_{i=1}^{n_i} P_i$$

$$; N_j = \prod_{j=1}^{n_j} P_j$$

$P_{Ri}$ referred to the probability that CDV use the entry pathway to wildlife stations from the outside for each introduction pathway. It represented six scenario trees of the entry pathway probabilities through i) CDV-infected tigers ($P_{Rt}$), ii) other CDV-infected wild animals ($P_{Ra}$), iii) domestic dog or cat reservoirs ($P_{Rd}$), iv) contaminated human hands ($P_{Rh}$), v) contaminated animal transport vehicles ($P_{Rv}$), and vi) contaminated equipment ($P_{Re}$). $R_i$ represents at-risk entry sub-pathways, and $N_j$ represents non risk entry sub-pathways (S2–S5 Figs in S1 File). $P_i$ and $P_j$ were the probabilities of each node in all entry pathways (S6 Tables in S1 File). Each probability of entry pathway ($P_{Ri}$) was estimated, and all six entry pathways were combined to quantify the risks of having all entry pathways of CDV infection and contamination pathways annually at the final step.

### Sensitivity analysis

The aforementioned model was subjected to sensitivity analysis using Spearman's rank correlation to assess the effect of uncertainty on the model and identify the most influential parameters. The correlation coefficients ($\rho_s$) between each entry pathway and the final entry probability of CDV introductions to captive tigers were calculated. A tornado graph was used to rank all inputs according to their contributions to the variance of the output. Sensitivity analysis can also help identify appropriate interventions to reduce CDV infection risk.

## Risk-mitigated interventions

After identifying the entry pathway with the most significant effect on the final probability, interventions to reduce the overall risk can be determined. Feasible interventions were incorporated into the biological pathways to assess the reduction in entry risk probabilities. By implementing these interventions and noting the reduction in CDV infection risk, efforts to minimize the risk effectively can be prioritized. The probability of entry with interventions ($P_{R_j}$) for CDV infection or external contamination to tigers in wildlife stations was modeled as follows:

$$P_{R_j} = \sum_{i=1}^{n_i} R_{Ii} \bigg/ \left( \sum_{i=1}^{n_i} R_{Ii} + \sum_{j=1}^{n_j} N_{Ij} \right)$$

$$; R_{Ii} = \prod_{i=1}^{n_i} P_{Ii}$$

$$; N_{Ij} = \prod_{j=1}^{n_j} P_{Ij}$$

The following are the probabilities of CDV introduction to captive tigers in wildlife stations with integrated risk-mitigated interventions ($P_{Rj}$) via six scenarios: $P_{Rt}$, $P_{Ra}$, $P_{Rd}$, $P_{Rh}$, $P_{Rv}$, and $P_{Re}$. $P_{Rjr}$ represented the routine intervention practices, and $P_{Rjf}$ described 100% or implementations of all interventions. The interventions implemented in all scenarios were as follows: i) screening or reverse-transcription polymerase chain reaction (RT-PCR) tests from feces, sick tiger isolation and treatment, and CDV vaccine efficacy and coverage ($P_{Rtr}$ and $P_{Rtf}$) for the probability of entry via CDV-infected tigers; ii) screening or RT-PCR tests from feces, sick animals isolation and treatment, CDV vaccine efficacy, and coverage for the probability of entry via other CDV-infected wild animals ($P_{Rar}$ and $P_{Raf}$); iii) preventing dogs or cats from entering near tiger cages for the probability of entry via domestic dog or cat reservoirs ($P_{Rdr}$ and $P_{Rdf}$); iv) thorough hand washing for the probability of entry via contaminated human hands ($P_{Rhr}$ and $P_{Rhf}$); v) effective disinfection for the probability of entry via contaminated animal transport vehicles ($P_{Rvr}$ and $P_{Rvf}$); and vi) not sharing equipment between animal cages for probability of entry via contaminated equipment ($P_{Rer}$ and $P_{Ref}$). $R_{Ii}$ represents at-risk entry sub-pathway with interventions, and $N_{Ij}$ represents a non-risk one with interventions (S2–S5 Figs in S1 File). $P_{Ii}$ and $P_{Ij}$ indicated the probabilities in each node in all entry pathways with the implementation of integrated risk-mitigated interventions (S6 Tables in S1 File). Regarding vaccination, the term "efficacy" refers to the reduction rate in disease incidence among vaccinated individuals compared with unvaccinated individuals, calculated as follows: VE = ((ARU-ARV)/ARU) * 100, where ARU is the attack rate in the unvaccinated population, and ARV is the attack rate in the vaccinated population [19]. "Coverage" refers to the proportion of the target population that has been vaccinated. In this study, vaccine coverage was calculated based on questionnaire data collected from wildlife stations, detailing the number of vaccinated tigers and other wildlife species.

The relative risk reduction of the risk probability in each entry pathway ($\Delta P_R$) was calculated by the risk probabilities with ($P_{R_j}$) and without ($P_{R_i}$) interventions using the following formula:

$$\Delta P_R = (P_{R_i} - P_{R_j}) / P_{R_i}$$

The effect of implementing a single intervention on the probability of CDV entry to captive tigers in each pathway was also analyzed. The input probabilities of CDV entry pathways with interventions varied from 0% to 100%, and the output risk probabilities of CDV entry were recorded in each scenario.

Many patterns of the risk-mitigated interventions were assessed for risk reductions on entry routes I and II. The interventions of sick tiger isolation and treatment ($P_{Rtiso}$), CDV tests performed at original stations ($P_{Rttor}$), destined stations ($P_{Rttst}$), and both original and destined wildlife stations ($P_{Rttos}$) and effective vaccinations ($P_{Rtvac}$) were added in the risk probability of entry of the tiger ($P_{Rt}$). The interventions for isolating and treating other sick wild animals ($P_{Raiso}$); performing CDV tests at original stations ($P_{Rator}$), destined stations ($P_{Ratst}$), and both original and destined wildlife stations ($P_{Ratos}$); and giving effective vaccinations ($P_{Ravac}$) were added in the risk probability of entry of other wild animals ($P_{Ra}$).

Information on captive tigers and other wild animals infected with CDV annually was derived from questionnaires and the health and husbandry records of DNP wildlife stations, which were integrated with the laboratory database of the Monitoring and Surveillance Center for Zoonotic Diseases in Wildlife and Exotic Animals of Mahidol University. Information on dog or cat reservoirs, contaminations through human hands, wheels or mudguards of animal transport vehicles, and equipment was derived from questionnaires and literature reviews (S6 Tables in S1 File).

### Final risk estimation

The risk probability in each entry node was assessed, six stochastic quantitative risk assessment models were calculated, and probabilities were combined to estimate the final risk probability of CDV introduction to captive tigers in Thailand's wildlife stations, with and without interventions annually. $P_{Rfin}$ was the final entry probability of CDV introduction to captive tigers in wildlife stations without interventions. $P_{Rfinr}$ referred to the final entry probability of CDV introduction to captive tigers in wildlife stations with all routine interventions. $P_{Rfinf}$ indicated the final entry probability of CDV introduction to captive tigers in wildlife stations with the implementation of all interventions (100%). The annual probabilities of CDV introduction from six entry pathways were independent events and were not mutually exclusive. Therefore, the combined probabilities of all CDV entry pathways with and without interventions were estimated as follows:

$$P_{Rfin} = 1 - \left[ \left(1 - P_{R_t}\right) * \left(1 - P_{R_a}\right) * \left(1 - P_{R_d}\right) * \left(1 - P_{R_h}\right) * \left(1 - P_{R_v}\right) * \left(1 - P_{R_e}\right) \right]$$

$$P_{Rfinr} = 1 - \left[ \left(1 - P_{R_{tr}}\right) * \left(1 - P_{R_{ar}}\right) * \left(1 - P_{R_{dr}}\right) * \left(1 - P_{R_{hr}}\right) * \left(1 - P_{R_{vr}}\right) * \left(1 - P_{R_{er}}\right) \right]$$

$$P_{Rfinf} = 1 - \left[ \left(1 - P_{R_{tf}}\right) * \left(1 - P_{R_{af}}\right) * \left(1 - P_{R_{df}}\right) * \left(1 - P_{R_{hf}}\right) * \left(1 - P_{R_{vf}}\right) * \left(1 - P_{R_{ef}}\right) \right]$$

The relative risk reduction of the risk probabilities of all entry pathways, when integrated with all routine interventions ($\Delta P_{Rfinr}$), was calculated by the risk probabilities with ($P_{Rfinr}$) and without ($P_{Rfin}$) routine interventions using the following formula:

$$\Delta P_{Rfinr} = (P_{Rfin} - P_{Rfinr}) / P_{Rfin}$$

The relative risk reduction of the risk probabilities of all entry pathways, when integrated with the implementation of all interventions (100%) ($\Delta P_{Rfinf}$), was calculated by the risk probabilities with ($P_{Rfinf}$) and without ($P_{Rfin}$) the implementation of all interventions (100%) using the following formula:

$$\Delta P_{Rfinf} = (P_{Rfin} - P_{Rfinf}) / P_{Rfin}$$

### Ethical considerations

This study obtained approval for ethical considerations involving human subjects from the Mahidol University - Central Institutional Review Board (COE No. MU-CIRB 2022/130.2308). In addition, permission was secured from the Department of National Parks, Wildlife, and Plant Conservation (DNP) to conduct research at captive wildlife stations in Thailand from 2022 to 2023.

## Results

### Characteristics of respondents and wildlife stations

The questionnaire interviews involved 25 respondents, including 3 veterinarians (12%), 4 wildlife station directors (16%), 15 tiger keepers (60%), and 3 other keepers of other susceptible wild animals (12%). All had over 1 year of work experience and aged < 30 (8%), 30-50 (76%), and > 50 years (16%). The male-to-female ratio was 4:1. Two stations had full-time veterinarians, and five had veterinary care consultants from regional veterinarians of DNP.

Four out of seven wildlife stations are located near tiger habitats in the forest, and six are near the village (within 1-5 km). The wildlife stations obtained tigers and other wild animals from other DNP wildlife stations (100%) and other sources, e.g., confiscated wild animals from illegal trades or private collections, abandoned or injured wild animals (86%), and wild animals transported to other DNP wildlife stations (57%). Besides tigers, wildlife stations also contained species susceptible to CDV, i.e., wild cats (Asiatic golden cats, *Catopuma temminckii*; fishing cats, *Prionailurus viverrinus*; leopard cats, *Prionailurus bengalensis*; clouded leopards, *Neofelis nebulosa*; leopards, *Panthera pardus*; marbled cats, *Pardofelis marmorata*; and jungle cats, *Felis chaus*), wild dogs (Asiatic jackals, *Canis aureus*; and dholes, *Cuon alpinus*), civets (small Indian civet, *Viverricula indica*; large Indian civet, *Viverra zibetha*; large-spotted civet, *Viverra megaspila*; common palm civet, *Paradoxurus hermaphroditus*; and masked palm civet, *Paguma larvata*), binturongs (*Arctictis binturong*), bears (Asiatic black bear, *Ursus thibetanus*; Malayan sun bear, *Helarctos malayanus*), and monkeys (crab-eating macaque, *Macaca fascicularis*; pig-tailed macaque, *Macaca nemestrina*; rhesus macaque, *Macaca mulatta*; and stump-tailed macaque, *Macaca arctoides*). All wildlife stations allowed the entry of outsiders, and the respondents observed the presence of domestic dogs or cats around these wildlife stations.

### Probability of CDV entry into captive wildlife stations via different pathways

In this study, the median probabilities of six entry pathways that can introduce CDV to captive wildlife stations without any interventions for risk mitigation were assessed (Fig 3). The pathway with the highest probability of introducing CDV was through CDV-contaminated human hands ($P_{Rh}$), with a median probability of 0.534 (range, 0.372-0.719, 5th-95th percentiles). The second and third highest probabilities were not much different through CDV-contaminated equipment ($P_{Re}$), with a median probability of 0.379 (range, 0.196-0.619, 5th-95th percentiles), and CDV-infected wild animals ($P_{Ra}$), with a median probability of 0.376 (range, 0.181-0.622, 5th-95th percentiles). Other pathways, in descending order of probability, were the entry of CDV-infected tigers ($P_{Rt}$) (median probability, 0.111; range, 0.070-0.164, 5th-95th percentiles), entry of dog or cat reservoirs ($P_{Rd}$) (median probability, 0.045; range, 0.018-0.090, 5th-95th percentiles), and entry of CDV-contaminated vehicles ($P_{Rv}$) (median probability, 0.005; range, 0.001-0.011, 5th-95th percentiles), respectively. The probability estimations in each node of all entry pathways are shown in S7 Table in S1 File.

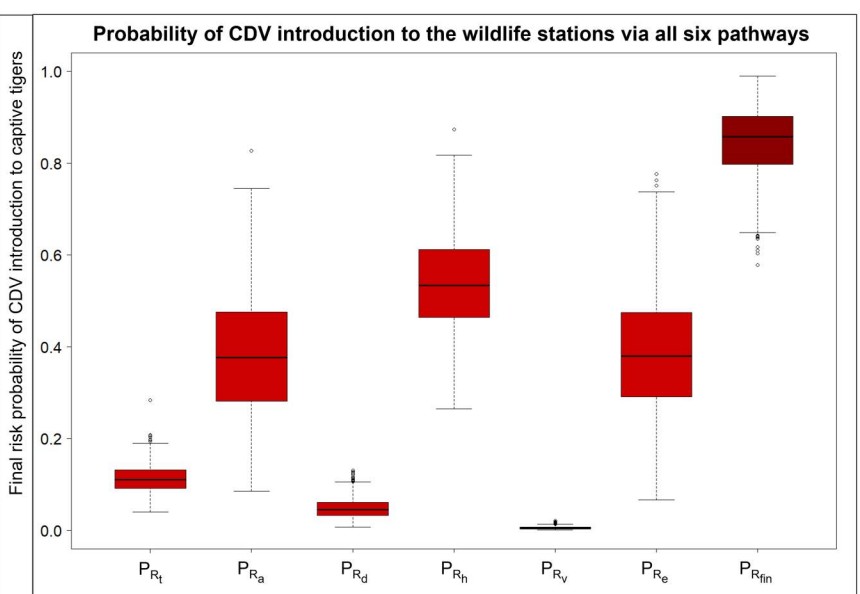

**Fig 3. Final probabilities of entry pathways of canine distemper infections and contaminations.** Box plots of each probability of entry ($P_{Rt}$, CDV-infected tigers; $P_{Ra}$, other CDV-infected wild animals; $P_{Rd}$, dog or cat reservoirs of CDV; $P_{Rh}$, CDV-contaminated human hands; $P_{Rv}$, CDV-contaminated animal transport vehicles; $P_{Re}$, CDV-contaminated equipment; and $P_{Rfin}$, final probability of entry without any interventions).

## Sensitivity analysis

The tornado graph (Fig 4A) illustrates the results of the sensitivity analyses. The correlation coefficients were assessed using Spearman's rank correlation. The most influential input for the final probability of CDV introduction to captive wildlife stations was the probability of entry via CDV-contaminated human hands ($P_{Rh}$), followed by other CDV-infected wild animals ($P_{Ra}$), CDV-contaminated equipment ($P_{Re}$), and CDV-infected tigers ($P_{Rt}$). The entry pathway with the smallest effect on the final probability of CDV introduction to captive wild-life stations was the probability of dog or cat reservoirs ($P_{Rd}$) and CDV-contaminated animal transport vehicles ($P_{Rv}$). Most of the inputs demonstrated a positive correlation, and only CDV-contaminated animal transport vehicles exhibited a negative correlation, with p-values < 0.05. Therefore, interventions consistent with the sensitivity analysis were selected, which mitigated the risk probabilities of entry (Fig 4B).

## Implementing of interventions

Fig 5 depicts the results of the effect of a single intervention on each scenario. The interventions were found to reduce the risk probabilities of each entry route in all pathways. The higher the percentage of interventions implemented, the higher the risk reductions. The implemented interventions were as follows: not allowing the entry of dogs or cats to wildlife stations, not sharing equipment between cages of tigers and other wild animals, performing CDV tests to tigers and other wild animals at both original and destined stations, hand washing, and disinfecting animal transport vehicles, which could relatively highly reduce the risk of entry in each pathway by 80-99%. Vaccinating animals without CDV testing slightly reduced the risk of entry in both tigers and other wild animals. Increasing CDV vaccine coverage in tigers and other wild animals reduced the risk more than increasing CDV vaccine efficacy.

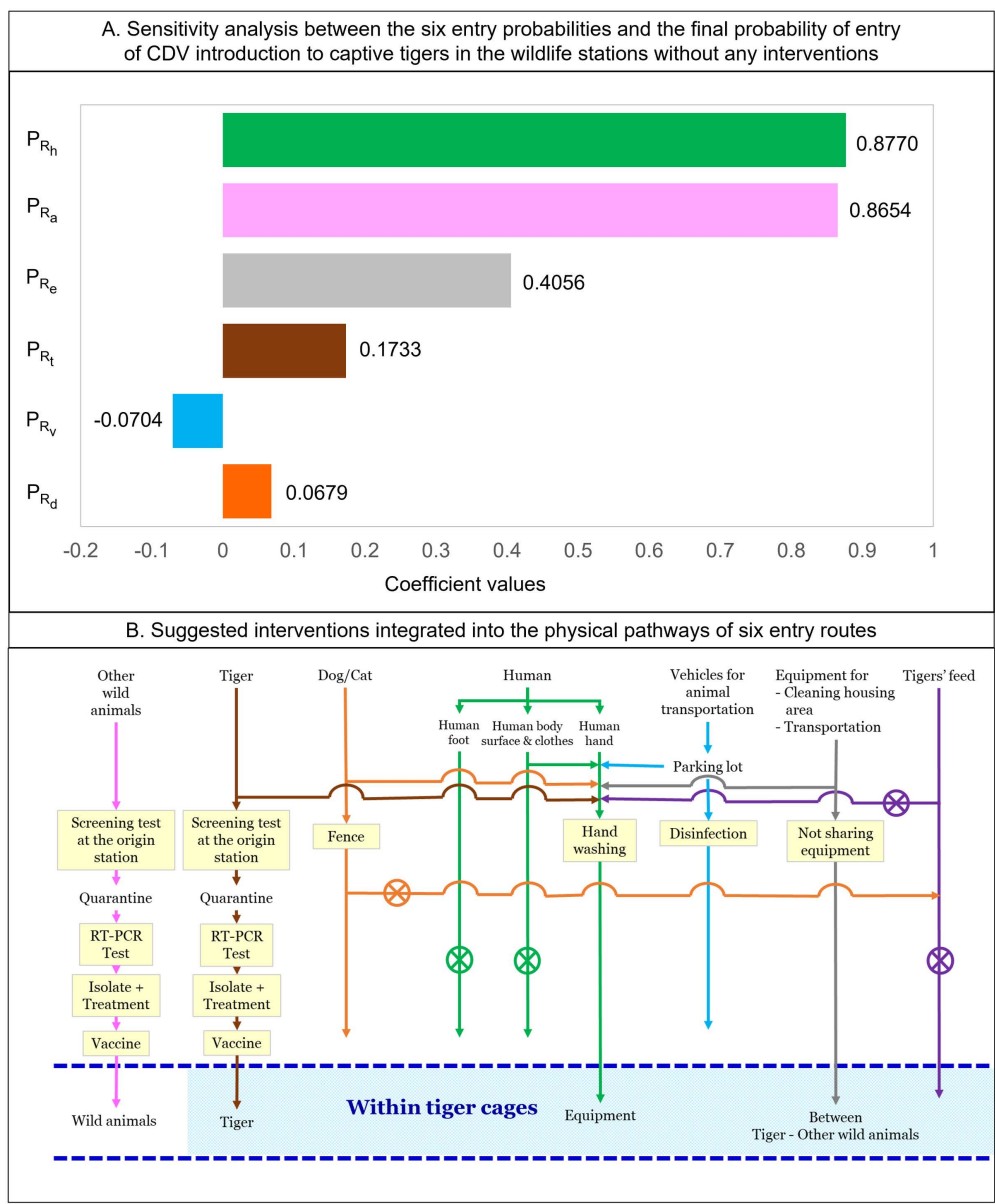

**Fig 4. Sensitivity analysis results of six entry pathways and physical entry pathways integrated with routine interventions.** (4A) Tornado graph depicting the sensitivity analysis of CDV risk introduction through infections and contaminations without interventions. Spearman's correlation coefficients ranked all input parameters according to their contributions to the output variance. ($P_{Rh}$, CDV-contaminated human hands; $P_{Ra}$, other CDV-infected wild animals; $P_{Re}$, CDV-contaminated equipment; $P_{Rt}$, CDV-infected tigers; $P_{Rv}$, CDV-contaminated animal transport vehicles; and $P_{Rd}$, dog or cat reservoirs of CDV). (4B) Physical pathways of humans, animals, and fomites entering wildlife stations with the selected interventions.

Fig 6 illustrates probabilities of entry and risk reductions when employing various patterns of risk-mitigated interventions in all six entry scenarios. In Fig 6A, routine (yellow box plots) testing of tigers for CDV before entry reduced the infection risk more than testing at their destination (median probability of 0.097 from 0.104). Fig 6B shows similar results for wild animals (median probability of 0.370 from 0.374). In both situations, the best way to minimize the risk is to perform CDV tests at the beginning and end of their

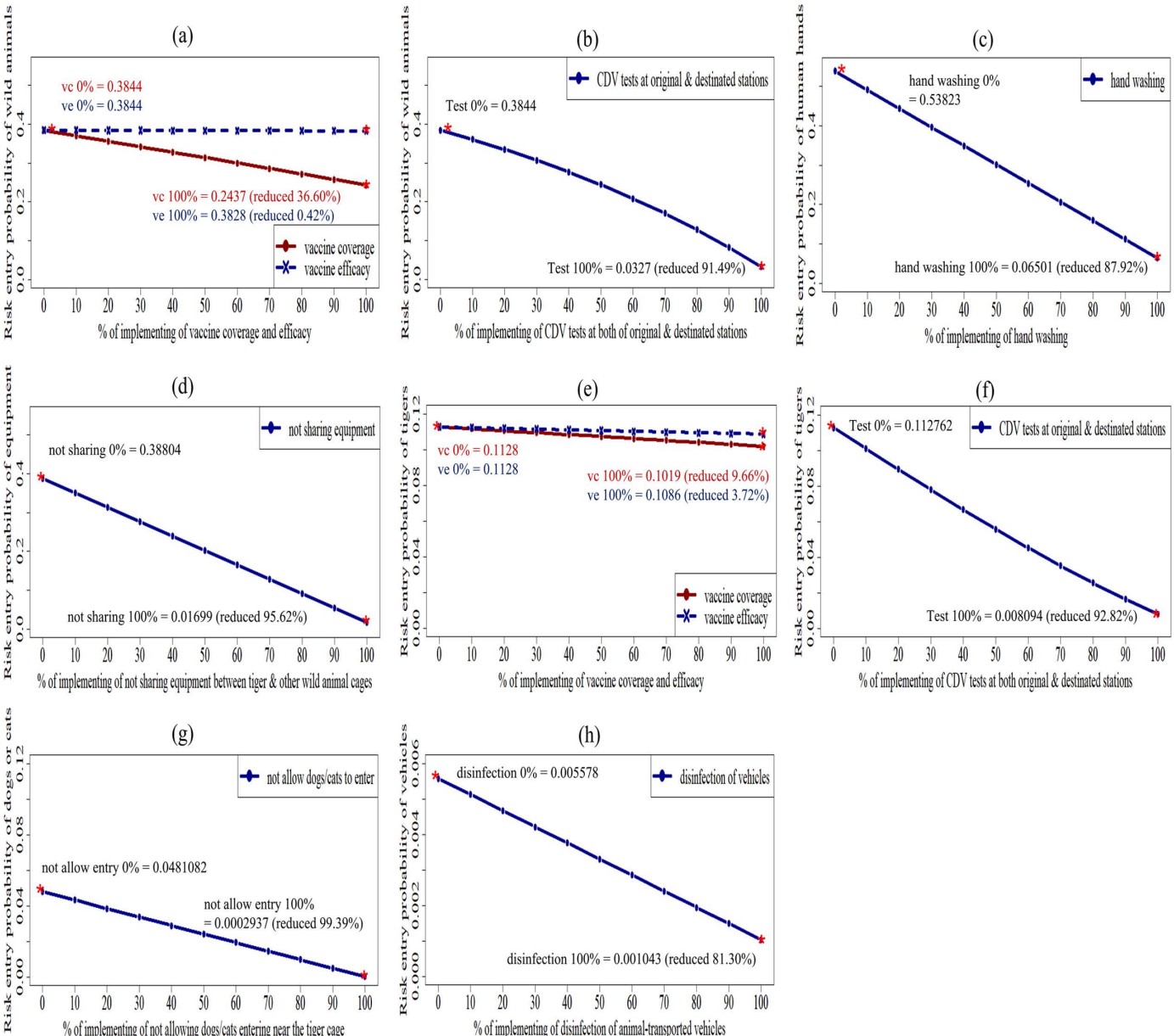

**Fig 5. Probabilities of risk reductions in each entry pathway using an input 0%-100% of risk-mitigated interventions.** (a) Vaccine coverage and efficacy in other wild animals. (b) CDV tests in other wild animals. (c) Hand washing of contaminated human hands. (d) No sharing equipment between tiger and wild animal cages. (e) Vaccine coverage and efficacy in tigers. (f) CDV tests in tigers. (g) Dogs or cats not entering areas near tiger cages. (h) Animal transport vehicle disinfection.

transportation (0.091, tigers; 0.368, wild animals). Testing at the origin is crucial. Animals that are tested positive should not be transported. Follow-up testing at the destination offers additional benefits, particularly if the animal was asymptomatic or a carrier during transport. This two-staged testing approach significantly enhanced the ability to identify and manage CDV risks effectively. A combination of testing, isolating, treating infected animals, and vaccinating healthy ones at entry points effectively reduced CDV spread in tigers and other wild animals. These routine combined interventions mainly reduced the

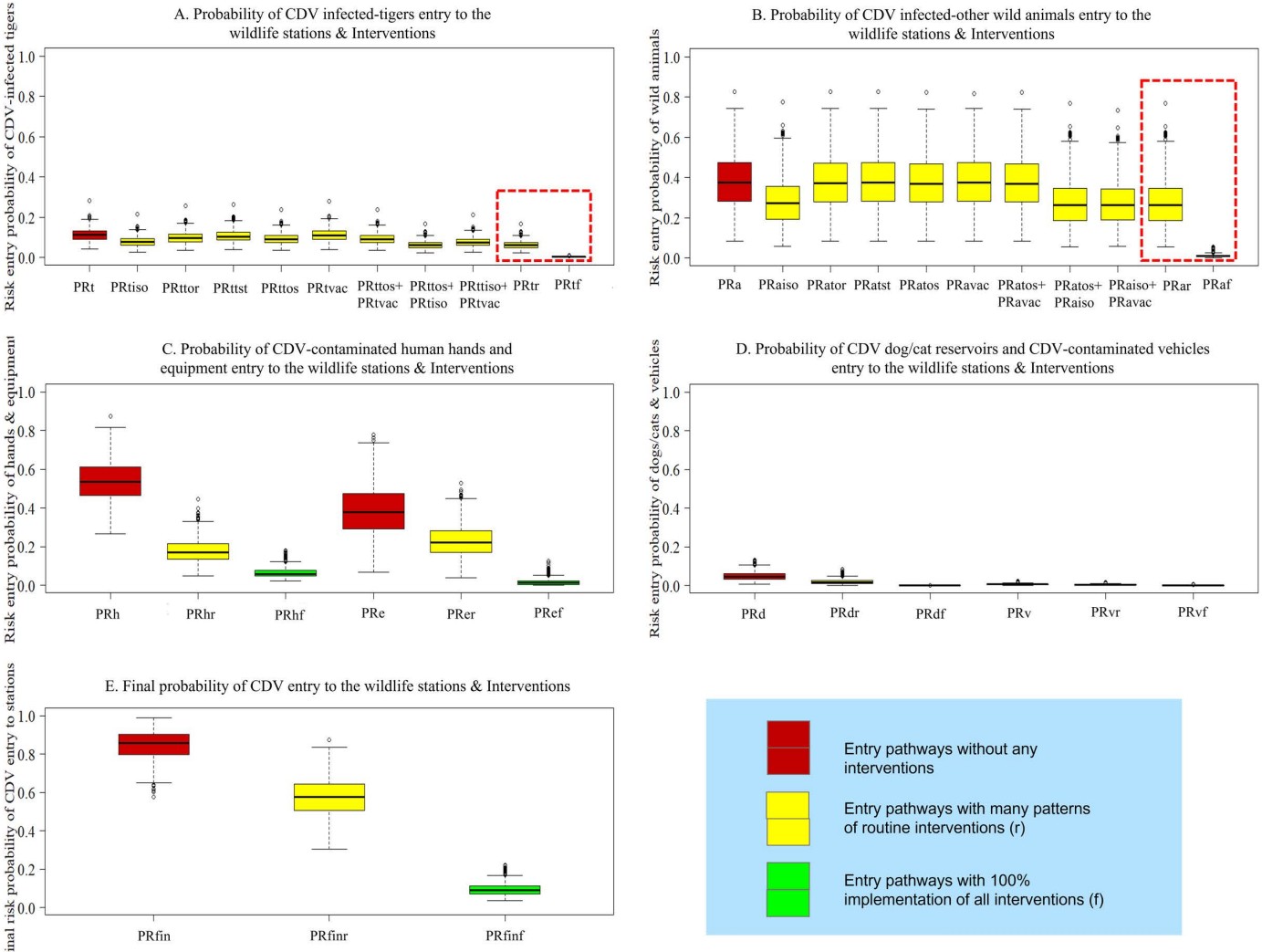

**Fig 6. Probabilities of entry and risk reductions using various risk-mitigated intervention patterns.** The red box plots indicate the conditions without interventions, yellow box plots with various patterns of routine interventions, and green box plots with 100% implementation of all interventions. (A) Entry probability through CDV-infected tigers. (B) Entry probability through other CDV-infected wild animals. (C) Entry probability through CDV-contaminated human hands and equipment. (D) Entry probability through dog or cat reservoirs and vehicles. (E) Final probability of CDV entry to wildlife stations.

risk of CDV transmission compared with other intervention patterns: $P_{Rtr}$ for tigers (0.060, reduced risk by 45.98%) and $P_{Rar}$ for wild animals (0.262, reduced risk by 30.37%). After fully implementing (100%) (green box plot) all interventions in tigers ($P_{Rtf}$), the risk was reduced from 0.111 to 2.420e$^{-03}$ (97.81%) compared with the initial risk ($P_{Rt}$). Similarly, when the same interventions were implemented in other wild animals ($P_{Raf}$), the risk was reduced from 0.376 to 0.009 (97.59%) (Fig 6A and 6B).

The risk reduction for contaminated human hands was achieved through routine thorough hand washing ($P_{Rhr}$), reducing the probability of CDV-contaminated human hands ($P_{Rh}$) from 0.534 to 0.169 (68.34%) (Fig 6C). When all humans in close contact with tigers thoroughly wash their hands (100%) ($P_{Rhf}$), the initial risk ($P_{Rh}$) decreased from 0.534 to 0.059, indicating a reduction of 88.96%. The risk of CDV-contaminated equipment ($P_{Re}$) was reduced by 41.54% (from 0.379 to 0.222) when equipment was not shared between tiger and other wild animal

cages in routine practice ($P_{Rer}$) and by 97.02% (from 0.379 to 0.011) when all equipment (100%) was not shared ($P_{Ref}$) (Fig 6C).

In Fig 6D, routine practice intervention reduced the risk associated with dog or cat reservoirs entering areas near tiger cages ($P_{Rdr}$) and wheels or mudguards of CDV-contaminated animal transport vehicles ($P_{Rvr}$), resulting in reductions of 66.88% (from 0.045 to 0.015) and 41.85% (from 0.005 to 0.003), respectively. When the entry of dog or cat reservoirs is avoided 100% ($P_{Rdf}$) and 100% of vehicles are disinfected ($P_{Rvf}$), the risk reduction rate is 99.42% (from 0.045 to $2.622e^{-04}$) and 84.25% (from 0.005 to $7.957e^{-04}$), respectively.

## Combined probabilities

As shown in Fig 6E, the final probability of entry of CDV introduction to the tigers in captive wildlife stations through animal infections and fomite contaminations, without any interventions ($P_{Rfin}$), was calculated from the entry probabilities of six pathways ($P_{Ri}$) in Fig 3, with a median probability of 0.858 (range, 0.716-0.950, 5th-95th percentiles). When utilizing routine or regular interventions, the final probability of entry of CDV introduction to the tigers in captive wildlife stations through animal infections and fomite contaminations ($P_{Rfinr}$) was indicated by a median probability of 0.578 (range, 0.416-0.746, 5th-95th percentiles). When implementing all integrated interventions (100%) in six scenarios, the final probability of CDV introduction to tigers in captive wildlife stations through animal infections and fomite contaminations ($P_{Rfinf}$) was reduced, with a median probability of 0.089 (range, 0.053-0.154, 5th-95th percentiles). Implementing routine practices for all interventions at entry points will result in a 32.60% reduction ($\Delta P_{Rfinr}$) in the risk of CDV entering a wildlife station. Fully implementing a comprehensive set of all interventions at entry points will result in an 89.57% reduction ($\Delta P_{Rfinf}$) in the risk of CDV entering a wildlife station.

The risks were reduced when routine practice and 100% implementation of all risk-mitigated interventions were employed (S8 Fig in S1 File). The probabilities of risks were compared between the entry pathways of implementing routine interventions in wildlife stations (yellow histogram), 100% of all interventions implemented (green histogram), and without any interventions (red histogram).

## Discussion

The entry of CDV-contaminated human hands poses the highest risk probability to captive tigers. Hand contamination can occur through direct contact with CDV-infected tigers, dog or cat reservoirs, contaminated equipment, or contaminated animal transport vehicles. Human hands can serve as contaminated fomites that transmit the virus by contact with body fluids and feces [20]. Routine thorough hand washing is important for preventing the spread of germs and can reduce the risk of CDV contamination by 68.34%. Washing hands thoroughly can reduce the risk of CDV contamination by 88.96%. This value is consistent with the results of a study reporting that 80% of germs can be transferred to hands during contact and can potentially contaminate objects or surfaces [21]. The use of plain or antiseptic soap effectively eliminates germs; even the use of plain soap and water alone can reduce bacteria by up to 92% [22–25]. Handwashing with soap and water or using alcohol-based hand rubs reduces the number of H1N1 virus particles to levels undetectable in cultures and approximately 100 virus copies/mL in PCR tests, achieving a reduction rate of > 99.99% in virus particles on hands [26]. H1N1 and CDV are negative-sense RNA viruses that share structural similarities, including lipid envelopes and surface proteins that are essential for entering host cells. They also have similar modes of transmission, including respiratory droplets, direct contact, and fomites [26]. These similarities make them highly susceptible to disinfectants that target lipid

envelopes, such as soap. Hand-washing facilities at farm entry and exit points reduced the likelihood of CDV outbreaks on Danish mink farms in 2012-2013 [27]. Handwashing can reduce respiratory illness risk by 6%-44% and prevent viral transmission, such as the measles virus, which is similar to CDV [24,28]. Alcohol-based hand rubs with 60% alcohol content are effective for hand hygiene when hands are not visibly dirty [28]. Campaigns should target staff and visitors entering wildlife stations to promote handwashing with plain soap. In addition, disposable gloves should be worn when handling tigers.

The introduction of other CDV-infected wild animals exhibited the second highest risk of CDV entry to captive tigers. Sensitivity analysis revealed that this pathway is more than twice as likely to contribute to CDV entry compared with contaminated equipment. Various interventions have been implemented to mitigate this risk, including CDV testing, isolating and treating sick animals, and ensuring effective vaccination protocols. Stress during animal transportation between stations can suppress the immune system, potentially triggering latent diseases such as CDV infection [29]. Performing CDV tests before entry to wildlife stations significantly reduces this risk compared with testing after entry. However, the best practice is to perform CDV testing both before and after entry. In addition, sourcing protocols for animals from exporting stations should include ensuring the absence of distemper in terrestrial carnivores within the previous 12 months and enforcing a 1-month pre-export and post-arrival quarantine at the importing station [18]. These measures, combined with ongoing CDV testing, are effective in monitoring and minimizing disease transmission risks [30].

The entry of CDV-contaminated equipment poses the third highest risk to captive tigers. Sharing equipment, such as floor-cleaning brushes or transport cages, can lead to the spread of CDV through contaminated surfaces. Strict protocols should be followed to reduce this risk, including avoiding the sharing of equipment between tiger and wild animal enclosures. Implementing such measures has decreased the risk of CDV contamination by 97.02%. This precaution aligns with earlier CDV outbreaks in farmed rhesus monkeys, captive African wild dogs, and farmed civets, where contaminated equipment or human handlers were recognized as potential indirect contact pathways of transmission [31–33]. Strict biosecurity measures are essential to mitigate this risk and safeguard the health of captive tigers.

The entry probability of CDV-infected tigers is the fourth highest risk for captive tigers and has less effect on the final probability of entry, following contaminated human hands, other CDV-infected wild animals, and contaminated equipment. This lower probability of entry and effect may be due to the relatively low prevalence of CDV in the healthy tiger populations in Thailand. If a tiger is infected with CDV at its place of origin, it must be treated until fully recovered before transport, which is already routine practice in normal situations. Although it is considered a lower priority, the strategies for mitigating this risk aligned closely with those employed in managing the entry of other CDV-infected wild animals. These strategies include performing CDV tests before and after entry, isolating and treating infected tigers, and administering effective vaccines. Quarantine at the importing station is crucial to mitigate the spread of CDV, a complex disease that affects large felines, often causing neurological symptoms and death. Clinical outcomes in infected tigers vary depending on their age and immune status, ranging from full recovery to persistent disease or fatality [34]. Isolating and treating symptomatic animals can significantly reduce CDV transmission risk [5,35,36]. Additionally, post-recovery viral shedding in dogs, which can occur in urine for 60–90 days [2,37] and in other secretions for up to 120 days, highlights the importance of extended quarantine and monitoring for complete resolution of the infection [38].

Implementing routine CDV vaccination if no CDV tests were performed in tigers and other wild animals slightly reduced the risk of entry. In contrast, routine CDV vaccination with previous CDV tests resulted in higher risk reduction. Moreover, CDV tests and isolation

and treatment of sick animals resulted in higher risk reduction than CDV testing with vaccination. These are attributed to the low efficacy and coverage of vaccines routinely used in wildlife stations. The vaccine efficacy was 20% in tigers compared with 88% in civets or other wild animals, and the vaccine coverage was 7% in tigers and 0% in other wild animals used in the models. Vaccine coverage is low because it is currently part of a trial aimed at determining whether the antibody titer can provide protection against CDV infection in tigers. Conversely, achieving 99% CDV vaccine efficacy and coverage in tigers and other wild animals, alongside previous CDV testing and isolation and treatment of sick animals, led to the highest risk reduction. In Thailand, tigers received recombinant CDV vaccines [5]. Recombinant vaccines are considered safer because they employ a canarypox vector to deliver antigens; however, multiple doses may be needed to establish lifelong immunity because they induce weaker immune responses [16]. Compared with recombinant vaccines, modified live or live-attenuated vaccines utilize either canine or avian kidney cell lines and elicit a significantly higher and extended humoral immune response [6]. However, vaccination with attenuated CDV strains was associated with an increased risk of post-vaccination CDV infection in untested species, highlighting the importance of thorough safety evaluations before administration [16,39]. Implementing test-based surveillance for infections in animals is a crucial strategy to minimize these risks. To ensure herd immunity, a percentage of the population must be vaccinated, which typically ranges from 60% to 90%; these figures are influenced by various population-specific factors [40–41]. The following factors must be considered in tiger or other wild animal vaccinations: population density, tigers' distance from other wild animals, prevalence of CDV infection, and host susceptibility. Vaccine efficacy is affected by the infectious properties of the pathogen, vaccine type, dose, administration route, host-specific factors (age, genetics, and immune health), and degree of similarity between the vaccine and the current circulating virus [30,42]. The most effective approach for preventing CDV transmission between domestic dogs and wild animals is to regularly vaccinate dogs and limit their contact with wild animals [34].

The entry of dog or cat reservoirs into wildlife stations poses a low risk of introducing CDV to captive tigers. This concern is heightened by the CDV situation in nearby villages. Wildlife stations close to these villages often have high populations of domestic dogs, which can exacerbate the virus spread. Increased domestic dog populations correlate with the likelihood of CDV transmission. Domestic dogs serve as the main reservoirs of CDV among wild animals, contributing to a decline in wild carnivore populations worldwide [1,43]. Implementing mass vaccination programs for dog populations near wildlife areas is recommended [1,44]. However, studies conducted in Africa have indicated that dog populations adjacent to wildlife areas may be inadequate to sustain the virus [1,45]. The ability of species to act as reservoirs depends on susceptibility, population size, turnover, and interaction frequency [16]. Although cats can become infected, they are generally asymptomatic. The virus spreads from dogs to cats but not vice versa or among cats [46]. In addition, no substantial evidence supports that CDV can be transmitted from cats to other animal species or wild animals. To restrict the entry of dogs and cats into wildlife stations and near tiger cages (within 6 m of aerosol transmission) [47], restrictions can be enforced, such as installing fencing, banning visitors from bringing pets, and relocating waste ponds away from the housing areas of wild animals.

Although the risk of entry of CDV-contaminated animal transport vehicles was very low to negligible, it still poses a risk. The duration of transportation is determined by distance. CDV can survive for extended periods at lower temperatures, such as up to 14 days at 5°C and 48 h at 25°C [20,48]. In a CDV culture study, the virus remained viable for up to 125 min at 25°C, but died faster at higher temperatures, lasting only 75 min at 35°C and 65 min at 37°C [25]. In Thailand, where temperatures fluctuate between 28 and 33°C in cooler months and between

30 and 37°C in the summer [49], the virus may survive during animal transport between nearby wildlife stations, which typically takes 1-2 h. The negative correlation between the entry of animal transport vehicles and the overall probability of CDV entry may stem from the fact that the distance over which animals are transported is inversely related to the ability of the virus to survive. However, factors such as heat, dry conditions, and disinfectants can effectively inactivate CDV [36]. Therefore, appropriate disinfection of animal transport vehicles is essential.

Lastly, the overall probability of CDV entry was calculated by integrating six scenarios. When no standard procedures or risk-mitigated interventions were implemented, the probability of CDV entry was relatively high. However, implementing all interventions across all pathways significantly reduced the risk of CDV entry and is recommended as a standard practice.

In addition, wildlife stations are situated near forests that are natural habitats of wild tigers and are mostly protected areas, including the Western Forest Complex, Kaeng Krachan Forest Complex, and the Eastern Forest [50]. This proximity poses a potential risk for the spread of CDV into the wild. Furthermore, the endangered Amur tiger faces extinction because of CDV infection [51]. Past CDV outbreaks have devastated various wildlife species, including silver-backed jackals (*Canis mesomelas*), bat-eared foxes (*Otocyon megalotis*), African wild dogs (*Lycaon pictus*), and Tanzania's lion (*Panthera leo*) in East Africa [52]. The presence of numerous susceptible species, disease reservoirs, and dense populations increases the risk of CDV transmission among nearby species.

Our processes are applicable to situations such as in our study setting. Conceptual and biological pathways must be reassessed for wildlife stations with varying animal husbandry and management practices. Nevertheless, this study has some potential limitations. For example, data's uncertainty may be caused by overestimating disease probabilities during all introductions. Specifically, the hand transmission rate of CDV was overestimated because of insufficient data regarding virus survival and transmission rate via hands. Conducting observational studies and research on mathematical models can help estimate relevant factors and unknown parameters, such as the probability of transmission, recovery, number of reservoirs, and vectors. In this study, the information was cross-checked with wildlife station records and animal test laboratory data to minimize recall bias of past events or behaviors during questionnaire interviews.

## Conclusions

In this study, the risk assessment models have identified critical points that need attention for captive wildlife conservation. We recommend implementing biosecurity measures, performing screening tests, providing appropriate vaccinations, and restricting human and animal movements into wildlife stations to prevent CDV outbreaks. Wildlife stations can utilize the results of quantitative risk assessments to improve their management and biosecurity protocols. Conducting an annual risk assessment is essential to monitor improvements in risk reductions and guide effective management practices. The health and well-being of tigers are crucial in species preservation and maintenance of ecological balance, whether they are in captivity or the wild.

## Supporting information

**S1 File. S1 Table 1. Captive tiger population number in Department of National Parks wildlife stations in 2021. S2 Figs 3A. Scenario trees depicting the biological pathways of CDV introductions into captive wildlife stations through infected tigers with and without**

**interventions.** (3A) Scenario tree without any interventions. (3.1A) With the intervention of isolation and treatment of sick tigers. (3.2.1A) With CDV testing at the place of origin. (3.2.2A) With CDV testing at the destined wildlife station. (3.2.3A) With CDV testing in both original and destined stations. (3.3A) With CDV vaccinations. (3.4A) With all combined interventions. (3.5A) With CDV testing in both stations and vaccinations. (3.6A) With CDV testing in both stations and isolation and treatment of sick tigers. (3.7A) With isolation and treatment of sick tigers and vaccinations. **S3 Figs 3B. Scenario trees depicting the biological pathways of CDV introductions into captive wildlife stations through other infected wild animals with and without interventions.** (3B) Scenario tree without any interventions. (3.1B) With isolation and treatment of sick wild animals. (3.2.1B) With CDV testing at the place of origin. (3.2.2B) With CDV testing at the destined wildlife station. (3.2.3B) With CDV testing in both original and destined stations. (3.3B) With CDV vaccinations. (3.4B) With all interventions. (3.5B) With CDV testing in both stations and vaccinations. (3.6B) With CDV testing in both stations and isolation and treatment of sick wild animals. (3.7B) With isolation and treatment of sick wild animals and vaccinations. **S4 Figs 3C. Scenario tree depicting the biological pathway of CDV introductions into captive wildlife stations through dog or cat reservoirs with and without interventions.** (3C) Scenario tree without interventions. (3.1C) With intervention of not allowing dogs or cats enter areas near tiger cages. **S5 Figs 3D-3F. Scenario trees depicting the biological pathways of CDV introductions into captive wildlife stations through contaminations of human hands, wheels or mudguards of animal transport vehicles, and equipment.** (3D) Scenario tree via contaminated human hands without interventions. (3.1D) With the intervention of thorough hand washing. (3E) Scenario tree via contaminated wheels or mudguards of animal transport vehicles without intervention. (3.1E) With the intervention of effective disinfection of vehicles. (3F) Scenario tree via contaminated equipment without intervention. (3.1F) With the intervention of not sharing equipment between tiger and other wild animal cages. **S6 Tables 2-4. Description of input parameters and probabilities utilized in the quantitative stochastic models for the entry assessment of the risk of CDV introduction into wildlife stations through tiger and other wild animal infections (Tables 2 and 3), dog or cat reservoirs, and contaminated hands, animal transport vehicles, and equipment (Table 4). S7 Table 5. Probability estimations in each node of all six entry pathways (number of iterations = 1,000). S8 Fig 7. Histograms showing risk reductions using interventions in different risk pathways.** The left red histograms show the condition without interventions, middle yellow histograms with routine interventions, and right green histograms with the implementation of all interventions (100%). Entry probabilities through (7a) CDV-infected tigers, (7b) CDV-infected wild animals, (7c) dog or cat reservoirs, (7d) CDV-contaminated human hands, (7e) CDV-contaminated animal transport vehicles, and (7f) CDV-contaminated equipment and (7g) final entry probabilities through all animal infection and fomite contamination routes. **S9. Questionnaires for interviews.** Questionnaire number 1 (QN1) was used for interviewing directors or veterinarians of wildlife stations. Questionnaire number 2 (QN2) was used for interviewing animal keepers of wildlife stations.
(PDF)

## Acknowledgments

We thank the Department of National Parks, Wildlife and Plant Conservation, Ministry of Natural Resources and Environment, Thailand, for allowing us to conduct research within their areas and providing the necessary information concerning both captive and non-captive tigers. We thank the Monitoring and Surveillance Center for Zoonotic Diseases in Wildlife

and Exotic Animals, Faculty of Veterinary Science, Mahidol University, Thailand, for providing the laboratory data on CDV in tigers and other wild animals and necessary information on CDV outbreak among captive tigers and allowing our study to proceed. We are grateful to the contributors for their kind cooperation during our questionnaire interviews and conducting the qualitative pathway assessment.

## Author contributions

**Conceptualization:** Kanittha Tonchiangsai, Anuwat Wiratsudakul, Suwicha Kasemsuwan, Ruangrat Buddhirongawatr, Sarin Suwanpakdee.

**Data curation:** Kanittha Tonchiangsai, Bencharong Sangkharak, Pakpoom Aramsirirujiwet.

**Formal analysis:** Kanittha Tonchiangsai, Sarin Suwanpakdee.

**Investigation:** Kanittha Tonchiangsai, Anuwat Wiratsudakul, Tatiyanuch Chamsai, Nareerat Sangkachai, Sarin Suwanpakdee.

**Methodology:** Kanittha Tonchiangsai, Anuwat Wiratsudakul, Suwicha Kasemsuwan, Ruangrat Buddhirongawatr, Kan Kledmanee, Sarin Suwanpakdee.

**Resources:** Bencharong Sangkharak, Pakpoom Aramsirirujiwet.

**Supervision:** Anuwat Wiratsudakul, Suwicha Kasemsuwan, Ruangrat Buddhirongawatr, Weerapong Thanapongtharm, Sarin Suwanpakdee.

**Validation:** Kanittha Tonchiangsai, Anuwat Wiratsudakul, Suwicha Kasemsuwan, Weerapong Thanapongtharm, Kan Kledmanee, Sarin Suwanpakdee.

**Visualization:** Kanittha Tonchiangsai, Anuwat Wiratsudakul, Suwicha Kasemsuwan, Tatiyanuch Chamsai, Sarin Suwanpakdee.

**Writing – original draft:** Kanittha Tonchiangsai, Anuwat Wiratsudakul, Suwicha Kasemsuwan, Ruangrat Buddhirongawatr, Sarin Suwanpakdee.

**Writing – review & editing:** Kanittha Tonchiangsai, Anuwat Wiratsudakul, Suwicha Kasemsuwan, Ruangrat Buddhirongawatr, Kan Kledmanee, Sarin Suwanpakdee.

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
