## [Decision Letter · Decision Letter 0]

9 Dec 2024

PONE-D-24-40378Quantitative risk assessment and interventional recommendations for preventing canine distemper virus in captive tigers at selected wildlife stations in ThailandPLOS ONE

Dear Dr. Suwanpakdee,

Thank you for submitting your manuscript to PLOS ONE. After careful consideration, we feel that it has merit but does not fully meet PLOS ONE’s publication criteria as it currently stands. Therefore, we invite you to submit a revised version of the manuscript that addresses the points raised during the review process. Please submit your revised manuscript by Jan 23 2025 11:59PM. If you will need more time than this to complete your revisions, please reply to this message or contact the journal office at plosone@plos.org . Please include the following items when submitting your revised manuscript:

We look forward to receiving your revised manuscript.

Kind regards,

Julian Ruiz-Saenz

Academic Editor

PLOS ONE

Journal Requirements: When submitting your revision, we need you to address these additional requirements. 1. Please ensure that your manuscript meets PLOS ONE's style requirements, including those for file naming. The PLOS ONE style templates can be found at https://journals.plos.org/plosone/s/file?id=wjVg/PLOSOne_formatting_sample_main_body.pdf and https://journals.plos.org/plosone/s/file?id=ba62/PLOSOne_formatting_sample_title_authors_affiliations.pdf

Reviewers' comments:

Reviewer's Responses to Questions

**Comments to the Author**

1. Is the manuscript technically sound, and do the data support the conclusions?

Reviewer #1: Yes

Reviewer #2: Partly

2. Has the statistical analysis been performed appropriately and rigorously? 

Reviewer #1: Yes

Reviewer #2: Yes

3. Have the authors made all data underlying the findings in their manuscript fully available?

Reviewer #1: Yes

Reviewer #2: Yes

4. Is the manuscript presented in an intelligible fashion and written in standard English?

Reviewer #1: Yes

Reviewer #2: No

5. Review Comments to the Author

**Reviewer #1: ** Dear authors,

I have carefully read the article entitled “Quantitative risk assessment and interventional recommendations for preventing canine distemper virus in captive tigers at selected wildlife stations in Thailand”. I find both the topic chosen and the comprehensive way of analysing each factor, together and separately, to be excellent.

Furthermore, I believe that the results of this article should be instituted as a comprehensive guide to prevent infections (not only CDV, but all pathogens using similar routes of transmission, and showing similar extra-animal resistance in enclosures where animals are kept).

In general:

1. The study presents the results of original research.

2. The reported results have not been published elsewhere.

3. Experiments, statistics and other analyses are carried out to a high technical standard and are described in sufficient detail, although there are minor issues that need to be clarified.

4. Conclusions are adequately presented and supported by data.

5. The article is presented in an intelligible form and is written in standard English.

6. The research complies with all applicable standards of experimental ethics and research integrity.

7. The article complies with appropriate reporting guidelines and EU standards on data availability, although some additional data should be presented as supplementary material.

Comments and suggestions:

Line 25: add the scientific name of the tiger.

Line 31: the phrase ‘ranging from 0.716 to 0.950 (5th to 95th percentile).’ seems unnecessary in the summary, as it does not provide much information as it does not contrast with other standard errors detected.

Line 34: in relation to ‘interventions’, perhaps it could be specified that they are interventions in an epidemiological context in zoos/captive breeding stations.

Line 39: in relation to ‘wildlife stations’, I would add zoos and private collections.

Line 41: the phrase ‘ ..... and it can be applied to protect even non-captive tiger populations.’ should be removed from this summary. It is not specified what these interventions in the natural habitat should be, nor is it referred to throughout the article.

Line 50: how is this probability found? is this an average probability? Explain or rephrase the sentence, as it gives the impression that all routes have the same probability of being entry routes.

Line 59: ‘can also be contracted’ would be better replaced by ‘could be transmitted through’.

Line 61: with regard to the phrase ‘the virus can even infect other carnivores like foxes, raccoons, bears, and large felids’, add the scientific names of the species that appear for the first time throughout the manuscript. Note that this virus can be detected in more than 60 species of species belonging to the Order Carnivora, and others outside this order.

Line 68: a reference to support this sentence is missing.

Line 69: a reference is missing to support this sentence.

Line 72: it would be appropriate to update the information in Table S1, excluding stations without tigers.

Line 72: scientific name of ‘white tigers’.

Line 73: Revise the sentence ‘For conservation value’.

Line 74: missing reference to support this sentence.

Line 76: missing reference to support this sentence.

Line 77: Put that it is in Siberia Amur Tigers (Pantera tigris sibirica). There is a missing reference to support this sentence.

Lines 77-79: I think these sentences should be reworded to explain more briefly the outbreak in Siberian tigers.

Lines 79-82: These sentences lack the proper context here. I would place them above where captive breeding centres are discussed,

Line 98: On ‘eighty eight’, I would choose between presenting the numerical information in word or in number.

Line 91: ‘suspected of CDV infection’ .... what is the prevalence?. There is a missing quotation for this sentence.

Line 98: add ‘captives’ before ‘large felids’.

Line 97-98: This sentence ‘The risk assessment results are helpful for the management of the quarantined animals, sick animal management, and viral particles spread into the environment.’ would be relocated to line 95, before ‘We used’.

Line 105: put 7 instead of seven.

Line 106: insert 2 instead of two.

Line 112: define ‘semi-structured’.

Lines 113-115: ‘We used various sources of information, including health and husbandry records of wildlife stations, laboratory databases, published literature, public databases, relevant websites, and standard guidelines.’ has this information been used to compile the survey? or what for? The survey used should be provided as supplementary material.

Line 129: ‘housing’ and ‘Cages’ (Fig 2) are used indiscriminately for the same thing. Define each word in order to be able to identify the spaces. Or use one of them.

Line 133: ‘CDV can enter to the wildlife stations through’.

Line 135: add ‘because not all transmission routes can infect tiger cages’ after ‘feed.’

Line 136: add ‘Following this guide,’ before ‘Tigers....’

Line 139: This information is confusing: ‘The presence of domestic dogs and cats around wildlife stations and their access to tigers or other wild animals’ cages were also considered in the entry pathway.’, as in Fig 2 this pathway does not appear to be able to enter cages.

Line 144: in relation to ‘tiger food’ is not considered as a netted pathway in Fig. 2.

Line 146: in relation to ‘food tongs’ is not considered as a netted pathway in Fig. 2.

Line 148: No comment on operators authorised to enter cages.

Line 149: Replace ‘cubs’ with ‘cages’. Add ‘external’ before ‘human’.

Line 156: In relation to what is stated in this paragraph, operators and outsiders are not allowed to contaminate the tigers' food?

Line 161: add after ‘equations’ ‘for transmission through tigers, other wild or domestic animals, fences and human hands are presented in’

Line 166: replace ‘six pathways through CDV infection or contamination from the outer to the destined wildlife’ by ‘six pathways through which CDV can infect or contaminate from the outer to the destined wildlife station’.

Line 171: perhaps it would be better to use ‘use the entry route to’ rather than ‘entered’.

Line 173: add ‘pathways’ after ‘entry’.

Line 184: perhaps in material and methods it would be interesting to homogenise the term ‘entry route’ by using either ‘entry pathway’ or ‘entry route’.

Line 187: add ‘CDV infection’ before ‘risk’.

Line 192: add ‘CDV infection’ before ‘risk’.

Line 203: define the terms ‘efficacy’ and ‘coverage’ in relation to vaccination.

Line 208: why have single-use gloves not been considered as a risk mitigation measure?

Line 211: why have protocols for proper disinfection of equipment not been considered as a risk mitigation measure?

Line 222: add ‘on entry routes I and II’.

Lines 277-280: add all scientific names of the detailed species.

Line 288: why only contaminated equipment has been considered and not infected wild animals? The results show that there are minimal differences between these two routes of entry, and furthermore, the rest of the routes are well below the first three considered.

Line 309: How do you explain the negative impact on the risk of transmission of animal transports?

Line 322: Figure 5 is poorly defined and it is very difficult to read the legends accompanying each graph.

Line 328-331: in relation to vaccination, the discussion should highlight that vaccination with attenuated strains of CDV has a high risk of post-vaccination CDV in some species, and that risk mitigation is much more effective with test-based surveillance for infection in animals.

Line 338: Missing definition in Figure 6, as in Figure 5.

Line 342: Can you explain better how it is possible to test an animal for CDV at the beginning of the journey without waiting for the result of the test? if it is positive it cannot travel....... Rephrase the sentence to better explain this.

Line 350: add (Fig 6A,B) at the end of the sentence.

Line 401: bibliographical references 14, 15, 16, 17, 20, 21 and 22 do not appear in the text of the Discussion. Reference 23 is incorrectly numbered, as it appears after reference 25.

References 18, 19, 23 and 24 are incorrectly numbered in the text as they appear after others with higher numbering.

Line 407: In addition, disposable gloves should be worn when handling tigers.

Line 408: I would suggest as above that CDV infected animals should be in second place of importance, as there is little difference with contaminated equipment, and furthermore, in the sensitive analysis it is more likely (almost twice as likely) to be a route of entry than contaminated equipment.

Line 414: Reference 31 is on rhesus monkeys, and 33 on civets in China, so it must be a mistake that they are referencing a case in African wild dogs. Check.

Line 416: I think it would be the second most important net route. If it is not, I would like a coherent justification.

Lines 428-434: this paragraph is out of context here. It would be an introductory paragraph to the discussion. Also, references 18 and 19 are misnumbered in this position.

Line 458: reference 39 is incorrectly numbered.

Line 483: reference 24 is incorrectly numbered.

Lines 497-498: add the scientific names of species that have not appeared in the text.

Line 575: this reference is easy to find? if not, it should be replaced by another accessible one.

Line 656: I could not find reference 39 in the text.

-------

**Reviewer #2:**  The manuscript addresses a critical and practical topic by examining the risk of canine distemper virus (CDV) infection in tiger populations, which is a key concern for the conservation of both captive and non-captive large felids. CDV poses a significant threat to these populations due to its high morbidity and mortality rates, making the study highly relevant to wildlife management and veterinary epidemiology. The authors’ focus on assessing CDV risk pathways and evaluating the effectiveness of combined preventive measures provides valuable insights for the development of targeted interventions to mitigate disease spread. The practical applications of the findings, particularly in adapting standard operating procedures for wildlife stations, have the potential to significantly contribute to tiger conservation efforts globally.

However, while the topic is well-chosen and the manuscript offers noteworthy findings, I have some questions regarding the methodology and the clarity of certain sections of the study. Specifically, the definition and explanation of the entry pathways require further elaboration to ensure the model's outputs are fully comprehensible and reproducible. Additionally, I recommend that the manuscript undergoes a thorough review by a native English speaker or professional editor to address some language inconsistencies and enhance overall readability.

Given these points, I recommend the manuscript for major revision. Below, I outline my detailed comments, highlighting areas that require further elaboration and improvement.

Major comments:

1. It would be beneficial to include the questionnaire used in the interviews as supplementary material. Providing access to the full questionnaire would enhance transparency and offer a clearer understanding of the data collection process and its structure.

2. The rationale behind the selection of specific categories in the analysis requires further clarification. Why were these particular categories chosen, and how do they align with the risk assessment model's assumptions?

- Line 158-160: i) CDV-infected tigers, ii) CDV-infected other wild animals, iii) dog or cat reservoirs, iv) CDV-contaminated human hands, v) animal-transported vehicles’ wheels or mudguards, and vi) equipment contaminated with CDV

2.1. Were all wild animals CDV-infected in the study (or “susceptible to CDV” Line 276)? Were there any non-infected wild animals that came into or could have come into contact with the tigers?

2.2. Regarding pathway ii), which wild animals are specifically included in this category? I recommend listing these animals along with their scientific names for clarity.

2.3. What is the practical and biological difference between categories i) and ii)? Why were wild animals and the domestic dog/cat categories separated into distinct biological pathways?

2.4. I have a similar question regarding pathways v) and vi). What is the practical and biological distinction between “v) animal-transported vehicles’ wheels or mudguards” and “vi) equipment contaminated with CDV”?

Minor comments:

Introduction

Line 98: I suggest expanding the sentence as follows: “and preventing the spread of viral particles into the environment.”

Materials and methods

Line 105: I suggest expanding the sentence as follows: “seven station”

Line 112-117: I understand that the data will be presented in detail later; however, to improve transparency, the "Data Collection" section could be expanded. For instance, it would be helpful to include details such as the total number of interviews conducted and the distribution of completed interviews across different stations. This additional information would provide a clearer understanding of the methodology.

Line 135, 144, 146, 155: tiger food or feed?

Results

Line 276-279: I recommend providing the precise identification of species, including their scientific names, to ensure clarity and accuracy.

Line 298: Were the dogs and cats also infected with CDV?

Discussion

Line 399-401: I suggest replacing the bacterial examples with studies focused on viruses. If the bacterial references remain in the text, I recommend revising the phrase '80% of germs can get on your hands and then go on to other things' to something like: 'can be transferred to hands during contact and potentially contaminate objects or surfaces’

Line 496-498: I suggest including the scientific names of the species that have not been mentioned previously in the text.

6. PLOS authors have the option to publish the peer review history of their article (what does this mean? ). If published, this will include your full peer review and any attached files.

**Do you want your identity to be public for this peer review?** For information about this choice, including consent withdrawal, please see our Privacy Policy .

Reviewer #1: No

Reviewer #2: No

---

## [Author Response · Author response to Decision Letter 1]

2 Feb 2025

Response to reviewers

We appreciate your careful and thorough review of our manuscript. Your insightful comments and recommendations have greatly enhanced its clarity and quality.

In this study, our quantitative risk assessment employed a stochastic model to estimate the probability of CDV introduction to captive tigers in the DNP wildlife stations through potential pathways of infection and contamination. Based on your valuable feedback, we have revised and refined the manuscript to ensure it is more robust and better aligned with your suggestions. In addition, we have refined the manuscript's writing with help from a native English speaker by Enago to enhance clarity and readability. The certificate of language editing is attached for the revised submission.

#Reviewer 1:

Line 25: add the scientific name of the tiger.

Response: We have added the scientific name of the tiger “Panthera tigris” in Line 25.

Line 31: the phrase ‘ranging from 0.716 to 0.950 (5th to 95th percentile).’ seems unnecessary in the summary, as it does not provide much information as it does not contrast with other standard errors detected.

Response: We have removed it in Line 30.

Line 34: in relation to ‘interventions’, perhaps it could be specified that they are interventions in an epidemiological context in zoos/captive breeding stations.

Response: We have specified the interventions in an epidemiological context that were used in captive wildlife stations in Lines 33-43 as follows:

Risk probabilities were compared among those without intervention, with routine intervention at wildlife stations, and with full intervention implementation. Implementing all interventions at the captive wildlife stations significantly reduced the risk of CDV introduction. These interventions included control measures such as quarantining and isolating infected animals and providing treatment to reduce infectiousness. Preventive measures included screening tests for healthy individuals for early detection of asymptomatic or pre-symptomatic cases, preventing further spread or complications, CDV vaccination campaigns, and promoting hand hygiene among staff and visitors. Environmental interventions involve restricting dogs and cats from accessing tiger enclosures, disinfecting animal transport vehicles, using separate equipment for each cage, etc. Together, these interventions lowered the median risk of CDV introduction to 0.089, representing an 89.6% risk reduction.

Line 39: in relation to ‘wildlife stations’, I would add zoos and private collections.

Response: We would like to clarify why we used “wildlife stations”: In the context of Thailand, there are clear differences in the management of wildlife stations, zoos, and private collections. The DNP wildlife stations where we collected data consist of wildlife breeding centers and wildlife rescue stations, which have similar management practices. These stations emphasize maintaining natural-like environments for the animals, minimizing human interaction to reduce stress, and ensuring the animals’ welfare. Limited public access is maintained to avoid disturbance. Zoos, in contrast, focus on creating safe and enriching environments for animals while also providing educational experiences for the public. They are typically open to visitors, often with an entrance fee, and their operations are often geared towards education, recreation, and conservation awareness. Private collections, which require permits from the DNP and must comply with wildlife laws, vary greatly in their management depending on the owner’s goals. While they may serve conservation purposes, they often face challenges in providing sufficient expertise, resources, or facilities for the animals, which can impact the overall welfare of the wildlife. Management in private collections is more variable compared to wildlife stations or zoos, as it is largely determined by the individual owner’s objectives and capacity. Our study focused only on wildlife breeding centers and rescue stations, not zoos and private sectors.

Additionally, we added the following sentences that informed the different types of wildlife stations, zoo, and private sectors: “Meanwhile, captive tigers are housed in wildlife stations and zoos and by private owners [9-11]. The government has implemented strict laws to protect tigers in Thailand [12]. The Department of National Parks, Wildlife, and Plant Conservation (DNP) oversees wildlife breeding and rescue centers (referred to as wildlife stations in this study), and the Zoological Park Organization of Thailand (ZPOT) manages governmental zoos.” Please see Lines 75-80.

Line 41: the phrase ‘ ..... and it can be applied to protect even non-captive tiger populations.’ should be removed from this summary. It is not specified what these interventions in the natural habitat should be, nor is it referred to throughout the article.

Response: We have removed this phrase, and we also made the sentences flow more smoothly in Lines 43-46 as follows:

This approach assessed CDV infection risks and adapted interventions to specific situations at wildlife stations. Consistent implementation of these measures is essential to minimize CDV spread. Wildlife stations must strictly implement these interventions as standard procedures to protect the health of captive tigers.

Line 50: how is this probability found? is this an average probability? Explain or rephrase the sentence, as it gives the impression that all routes have the same probability of being entry routes.

Response: We have rephrased the sentence in Line 54 as follows: "The results revealed that the virus could enter wildlife stations with a high probability of 85.8%, representing the combined risk probability across all entry routes.”

Line 59: ‘can also be contracted’ would be better replaced by ‘could be transmitted through’.

Response: We have changed the phrase in Lines 63-65 to “CDV is spread through the air by infected coughs or sneezes and could be transmitted through contact with bodily fluids [2].”

Line 61: with regard to the phrase ‘the virus can even infect other carnivores like foxes, raccoons, bears, and large felids’, add the scientific names of the species that appear for the first time throughout the manuscript. Note that this virus can be detected in more than 60 species of species belonging to the Order Carnivora, and others outside this order.

Response: We have added the scientific names of the species and rephrased the sentence in Lines 66-69. Although puppies and unvaccinated dogs are most at risk, all dogs are susceptible to the virus [2]. It can also infect other animals within the order Carnivora, including foxes (Vulpes vulpes), raccoons (Procyon lotor), bears (Ursus spp.), and large felids (Panthera spp.) and species outside this order, such as the rhesus monkey (Macaca mulata), collared anteater (Tamandua tetradactyla), and Asian elephant (Elephas maximus) [2].

Line 68: a reference to support this sentence is missing.

Response: We have added the references in Line 76 to support this sentence as “Meanwhile, captive tigers are housed in wildlife stations and zoos and by private owners [9-11].”

9. Wildlife Conservation Office, Department of National Park, Wildlife and Plant Conservation (DNP). National Action Plan for Tiger Conservation [Internet]. 2022 [cited 2024 Dec 18]. Available from: https://portal.dnp.go.th/DNP/FileSystem/download?uuid=5e0f2d2e-b49a-4395-92b2-a7166579f535.pdf.

10. The Zoological Park Organization, Thailand. Zoo animal population online database [Internet]. 2020 [cited 2021 Oct 21]. Available from: https://www.zoothailand-service.com/quantity/population/1.

11. Environmental Investigation Agency. Cultivating demand: The growing threat of tiger farms [Internet]. 2017 [cited 2024 Dec 18]. Available from: https://eia-international.org/report/cultivating-demand-growing-threat-tiger-farms/.

Line 69: a reference is missing to support this sentence.

Response: We have added a reference in Line 77 to support this sentence as “The government has implemented strict laws to protect tigers in Thailand [12].”

12. Wildlife Conservation Office, Department of National Park, Wildlife and Plant Conservation (DNP). Ministerial Regulation designating certain types of wild animals as protected animals, B.E. 2024 [Internet]. 2019 [cited 2024 Dec 18]. Available from: https://portal.dnp.go.th/DNP/FileSystem/download?uuid=8a059247-3011-4aaa-bbd8-39bbb9b337ef.pdf.

Line 72: it would be appropriate to update the information in Table S1, excluding stations without tigers.

Response: We have excluded the wildlife stations without tigers, as updated in Table S1.

Line 72: scientific name of ‘white tigers’.

Response: White tigers from zoos are the Bengal tigers (Panthera tigris tigris). The white tiger has a distinctive white fur color caused by a pigment abnormality. It is often found in the Bengal tiger, whose fur is usually orange. Thus, we decided to include them (seven white tigers) in the total 45 tigers in ZPO zoos (Please see Line 81). We also made it more apparent that the Zoological Park Organization of Thailand (ZPOT) manages governmental zoos (Please see Lines 79-80).

Line 73: Revise the sentence ‘For conservation value’.

Response: We revised and rewrote these sentences to “In tropical and subtropical ecosystems, tigers serve as an umbrella and iconic species, and their conservation supports broader biodiversity by maintaining ecological balance and protecting coexisting species [13]. As apex predators, they regulate prey populations, contributing to their ecosystem health [13].” Please see Lines 84-87.

Line 74: missing reference to support this sentence.

Response: We have added and changed the reference number to 13 in Line 86 and 14 in Line 87.

13. Dhakal M and Baral H. Tiger conservation in South Asia: lessons from Teari Arc landscapes, Nepal. Paper presented at the 2nd International Conference on Tropical Biology “Ecological restoration in Southeast Asia: challenges, gains, and future directions.” Bogor-Indonesia. October 12-13, 2015. 1-14.

14. Smithsonian's National Zoo & Conservation Biology Institute. Tiger: fact sheet [Internet]. [Cited 2025 Jan 21]. Available from: https://nationalzoo.si.edu/animals/tiger.

Line 76: missing reference to support this sentence.

Response: We have added a reference number 15 in Line 88 as follows:

However, wild tigers are facing an additional threat from CDV, which is associated with illness and death in large felids [15].

15. Bodgener J, Sadaula A, Thapa PJ, Shrestha BK, Gairhe KP, Subedi S, et al. Canine distemper virus in tigers (Panthera tigris) and leopards (P. pardus) in Nepal. Pathogens. 2023;12(2):203.

Line 77: Put that it is in Siberia Amur Tigers (Pantera tigris sibirica). There is a missing reference to support this sentence.

Response: We have specified species of tigers in Lines 88-90 and added a reference as follows:

CDV poses a potential threat to the extinction of Amur tigers (Panthera tigris altaica) with small populations, particularly those in Russia [14].

Lines 77-79: I think these sentences should be reworded to explain more briefly the outbreak in Siberian tigers.

Response: We have reworded these sentences in Lines 90-95 to “A significant CDV outbreak affected Amur tigers in 2004 and 2010 within the Sikhote-Alin Biosphere Reserve in Russia, leading to a dramatic population decline from 25 individuals in 2008 to only 9 in 2012. Infected tigers exhibited abnormal behaviors, such as fearlessness and neurological symptoms, which caused human-wildlife conflicts and fatalities. This outbreak highlighted the susceptibility of small tiger populations to diseases introduced from reservoirs of more abundant species, such as domestic dogs and wild carnivores [14].”

Lines 79-82: These sentences lack the proper context here. I would place them above where captive breeding centres are discussed,

Response: We have removed them to the end of the paragraph where captive wildlife stations are discussed. Please see Lines 81-83.

Line 90: On ‘eighty eight’, I would choose between presenting the numerical information in word or in number.

Response: We used the numerical information in number in Lines 102-104 as follows:

“Most of the 156 affected tigers were Siberian tigers taken from private collectors. A total of 88 tigers died; the majority of deaths were suspected to be caused by CDV infection, with 31.88% (22 out of 69 dead tested tigers) primarily attributed to CDV infection.”

Line 91: ‘suspected of CDV infection’ .... what is the prevalence?. There is a missing quotation for this sentence.

Response: We have added the suspected CDV infection prevalence among deaths in Line 104 as follows:

“A total of 88 tigers died; the majority of deaths were suspected to be caused by CDV infection, with 31.88% (22 out of 69 dead tested tigers) primarily attributed to CDV infection.”

Line 94: add ‘captives’ before ‘large felids’.

Response: We have added “captives’ large felids” and removed “species” in Line 107.

Line 97-98: This sentence ‘The risk assessment results are helpful for the management of the quarantined animals, sick animal management, and viral particles spread into the environment.’ would be relocated to line 95, before ‘We used’.

Response: We have relocated this sentence to Lines 108-110.

Line 105: put 7 instead of seven.

Response: We have put 7 instead of seven in Line 118.

Line 106: insert 2 instead of two.

Response: We have put 2 instead of two in Line 119.

Line 112: define ‘semi-structured’.

Response: We have defined ‘semi-structured’ in Lines 126-129 as follows:

Data were collected from 2016 to 2021 through interviews using semi-structured questionnaires. These questionnaires contained a mix of predefined questions with specific response options (structured parts) alongside open-ended questions that allowed respondents to provide more detailed responses.

Lines 113-115: ‘We used various sources of information, including health and husbandry records of wildlife stations, laboratory databases, published literature, public databases, relevant websites, and standard guidelines.’ has this information been used to compile the survey? or what for? The survey used should be provided as supplementary material.

Response: The mentioned sources of information in Lines 137-141 (health and husbandry records, laboratory databases, published literature, public databases, relevant websites, and standard guidelines) were not utilized to compile the survey. Instead, they were used as the input parameters and probabilities incorporated into the quantitative stochastic models employed in this study (please see Supplementary S6, Tables 2-4). Therefore, we would change the sentences as follows:

Nevertheless, information from various other sources, such as health and husbandry records of wildlife stations, laboratory databases, published literature, public databases, relevant websites, and standard guidelines, were used as input parameters and probabilities incorporated into the quantitative stochastic models employed in this study.

We also replaced the sentence “The interviews were conducted on-site and online.” after defining semi-structured questionnaires in

Line 129 to enhance flow and readability. We have provided the questionnaire survey that was approved by Mahidol University Central Institutional Review Board (MU-CIRB) in supplementary S9.

Line 129: ‘housing’ and ‘Cages’ (Fig 2) are used indiscriminately for the same thing. Define each word in order to be able to identify the spaces. Or use one of them.

Response: We chose to use housing areas instead of cages in Line 153 because they mean the facilities designed to provide shelter, protection, and a suitable environment for wildlife species which allow them to exhibit natural behaviors. Thus, we changed the words ‘Cages’ in Figs 2 and 4 to ‘housing areas’.

Line 133: ‘CDV can enter to the wildlife stations through’.

Response: We have added ‘wildlife stations’ after ‘CDV can enter’ in Line 159.

Line 135: add ‘because not all transmission routes can infect tiger cages’ after ‘feed.’

Response: We have added this phrase and refined the sentences in Lines 158-161 to as follows:

“Given that not all transmis

---

## [Decision Letter · Decision Letter 1]

24 Feb 2025

Quantitative risk assessment and interventional recommendations for preventing canine distemper virus infection in captive tigers at selected wildlife stations in Thailand

PONE-D-24-40378R1

Dear Dr. Suwanpakdee,

We’re pleased to inform you that your manuscript has been judged scientifically suitable for publication and will be formally accepted for publication once it meets all outstanding technical requirements.

Kind regards,

Julian Ruiz-Saenz

Academic Editor

PLOS ONE

Additional Editor Comments (optional):

Reviewers' comments:

Reviewer's Responses to Questions

**Comments to the Author**

1. If the authors have adequately addressed your comments raised in a previous round of review and you feel that this manuscript is now acceptable for publication, you may indicate that here to bypass the “Comments to the Author” section, enter your conflict of interest statement in the “Confidential to Editor” section, and submit your "Accept" recommendation.

Reviewer #1: All comments have been addressed

Reviewer #2: All comments have been addressed

2. Is the manuscript technically sound, and do the data support the conclusions?

Reviewer #1: Yes

Reviewer #2: Yes

3. Has the statistical analysis been performed appropriately and rigorously? 

Reviewer #1: Yes

Reviewer #2: Yes

4. Have the authors made all data underlying the findings in their manuscript fully available?

Reviewer #1: Yes

Reviewer #2: Yes

5. Is the manuscript presented in an intelligible fashion and written in standard English?

Reviewer #1: Yes

Reviewer #2: Yes

6. Review Comments to the Author

Reviewer #1: Dear authors,

Thank you for your comments and answers. The manuscript is really much more comprehensible.

I just think that Figures 4 to 6 are still not properly resolved.

I guess this is more editing than revision.

For my part, I think it is ready to be published (improving the quality of these figures), and that it will be very valuable information for any wildlife reserve, whether it is more or less naturalised.

Thanks for the effort.

Sincerely,

Mónica

Reviewer #2: (No Response)

7. PLOS authors have the option to publish the peer review history of their article (what does this mean? ). If published, this will include your full peer review and any attached files.

**Do you want your identity to be public for this peer review?** For information about this choice, including consent withdrawal, please see our Privacy Policy .

Reviewer #1: No

Reviewer #2: No

---

## [Editor Report · Acceptance letter]

PONE-D-24-40378R1

PLOS ONE

Dear Dr. Suwanpakdee,

I'm pleased to inform you that your manuscript has been deemed suitable for publication in PLOS ONE. Congratulations! Your manuscript is now being handed over to our production team.

Kind regards,

on behalf of

Dr. Julian Ruiz-Saenz

Academic Editor

PLOS ONE